# Evaluation of the Antifungal Activities of *Photorhabdus akhurstii* and Its Secondary Metabolites against Phytopathogenic *Colletotrichum gloeosporioides*

**DOI:** 10.3390/jof8040403

**Published:** 2022-04-15

**Authors:** Po-Wen Tu, Jie-Siang Chiu, Chih Lin, Chih-Cheng Chien, Feng-Chia Hsieh, Ming-Che Shih, Yu-Liang Yang

**Affiliations:** 1Agricultural Biotechnology Research Center, Academia Sinica, Taipei 11529, Taiwan; ray0983631720@gmail.com (P.-W.T.); chih@macrogen-europe.com (C.L.); ccchien1983@ntu.edu.tw (C.-C.C.); 2Biotechnology Center in Southern Taiwan, Academia Sinica, Tainan 71150, Taiwan; 3Institute of Plant Biology, College of Life Science, National Taiwan University, Taipei 10617, Taiwan; acerchu2002@yahoo.com.tw; 4Biopesticide Division, Taiwan Agricultural Chemicals and Toxic Substances Research Institute, Council of Agriculture, Taichung 41358, Taiwan; hsiehf@tactri.gov.tw

**Keywords:** *Photorhabdus akhurstii*, antifungal ability, glidobactin, biocontrol agent, natural product, *Colletotrichum gloeosporioides*, phytopathogen

## Abstract

*Colletotrichum gloeosporioides* is a phytopathogenic fungus that causes devastating losses in strawberries without effective countermeasures. Members of the genus *Photorhabdus* exhibit antimicrobial capability and have been found to have the potential for use as biocontrol agents against *C. gloeosporioides*. *Photorhabdus* species exhibit two phase variations with a differentiated composition of secondary metabolites designated to each phase. In this study, *Photorhabdus akhurstii* sp. nov. 0813-124 exhibited phase I (PL1) and phase II (PL2); however, only PL1 displayed distinct inhibition of *C. gloeosporioides* in the confrontation assay. We identified the bioactive ingredients of *P. akhurstii* sp. nov. 0813-124 to be glidobactin A and cepafungin I, with MIC values lower than 1.5 and 2.0 µg/mL, respectively. Furthermore, we revealed the biosynthetic gene cluster (BGC) of corresponding bioactive molecules through genomics analysis and determined its expression level in PL1 and PL2. The expression of glidobactin BGC in PL1 increased rapidly within 24 h, while PL2 was eventually stimulated after 60 h. In summary, we demonstrated that *P. akhurstii* sp. nov. 0813-124 could potentially be used as a biocontrol agent or part of a natural product repertoire for combating *C. gloeosporioides*.

## 1. Introduction

*Colletotrichum* spp. causing anthracnose disease results in severe decreases in strawberry production of about 50% or more [1,2,3]. Recent studies have revealed that *C. gloeosporioides* was one of the major fungal species causing anthracnose in major cultivation countries, e.g., China and America [2,4,5]. To date, the most common approaches employed to counter this disease have been using a high dose of chemical fungicide or antibiotics to repel fungal invasion [6]. However, this approach remains concerned with food safety, antibiotic residue, biocide resistance, and environmental contamination. Therefore, there is an urgent need to find an effective and sustainable strategy to act as a prospective alternative to counteract *C. gloeosporioides*.

The demand for biological controls, including microorganisms, metabolites, or natural products, is increasing rapidly [7]. To date, bacteria are the most developed natural product resource with comprehensive application in controlling pathogenic infection, such as *Pseudomonas, Bacillus,* and *Streptomyces*. Furthermore, innumerable bacteria are salient biocontrol agents against deleterious pathogens [8,9,10]. *Photorhabdus*, a genus of insect pathogenic Gram-negative bacteria belonging to the Enterobacteriaceae family, is an intestinal symbiont of the infective juvenile stage of *Heterorhabditis* nematodes. *Photorhabdus* are well known in the pesticide field [11]. Several studies have suggested that *Photorhabdus* contain plentiful non-ribosomal peptide synthase (NRPS) and polyketide synthase (PKS) genes encoding diverse types of bioactive secondary metabolites with antiparasitic and antimicrobial activities, such as benzylideneacetone, isopropylstilbene, mevalagmapeptide, phurealipid derivatives, phenethylamines, *trans*-cinnamic acid, and xenoamicin [12,13,14,15,16]. *Photorhabdus* species exhibit two phase variations with a differentiated composition of secondary metabolites designated primary and secondary phases [17,18]. During an extended cultivation period, a conversion of phenotype and phase unidirectionally occurs from primary variants into secondary variants with diverse phenotypic traits [19]. The two types of *Photorhabdus* are morphologically distinct: primary variants are long-rod shaped and can swim and swarm in aerobic or anaerobic conditions; the secondary variants are smaller short rods that barely move during aerobic incubation [20,21].

The primary variants yield more significant amounts of anthraquinones, stilbene, antibiotics, lipases, phospholipases, proteases, pigmentation, and bioluminescence than the secondary variants [14,21]. Although the secondary variants cannot produce anthraquinones and stilbene, the energy metabolism of secondary variants is more robust than those of the primary variants while growing in vitro and in vivo [14,22,23]. However, the physiological mechanism and the biological character of phenotypic variation are still vague.

Many articles have suggested that *Photorhabdus* species possess significant potential to be an ample resource to develop biological control agents. Notwithstanding, few studies have investigated the potency of the new species *P. akhurstii* against fungal pathogens. *P. akhurstii,* which was elevated from a subspecies of *P. luminescens* subsp. *akhurstii* to the species level [24], was recently demonstrated to possess antimicrobial capability through a bacterial extract from *P. akhurstii* that had considerable antibacterial effects on an antibiotic-resistant pathogen [25] and against selected post-harvest fungal phytopathogens [26]. The main objective of this study was to identify the major bioactive compounds produced by *P. akhurstii* that are responsible for the suppression of phytopathogenic fungus *C. gloeosporioides*. We also evaluated the antifungal capabilities in the two phase variants of *P. akhurstii*.

## 2. Materials and Methods

### 2.1. Microorganisms, Culture Media, and Growth Conditions

For bacteria, *P. akhurstii* sp. nov. 0813-124 (GeneBank Accession Number: DQ223040) with phase I (PL1) and II (PL2) isolated from entomopathogenic nematodes under the Heterorhabditidae family were provided by Dr. Feng-Chia Hsieh, Biopesticide Division, Taiwan Agricultural Chemicals and Toxic Substances Research Institute, Council of Agriculture [27]. The culture condition for *P. akhurstii* sp. nov. 0813-124 was slightly modified from Chen et al. [28]. Briefly, PL1 and PL2 were inoculated on iron-limited LP agar composed of M9 minimal medium, 1% casamino acids, 1 mM MgSO_4_, and 0.5% glycerol in a 30 °C incubator for 2 days.

For fungi, *C. gloeosporioides* was isolated from strawberries on a commercial farm in Taiwan and identified by the Taiwan Agricultural Chemicals and Toxic Substances Research Institute, Council of Agriculture, using microscopic morphological identification and fulfilling Koch’s postulates for *C. gloeosporioides*. Fungi were cultured on oatmeal agar and incubated at 30 °C for 7 days. After rinsing *C. gloeosporioides* growth plates with sterilized water, spore suspension was collected and filtered using sterilized Miracloth (Merck, Darmstadt, Germany) to remove mycelia. The filtrate was centrifuged at 4000× *g* for 10 min to harvest the spore pellet. The spore stock (10^8^ spores/mL) was obtained by resuspending the spore in sterilized double-distilled water and storing at room temperature for the following disk-diffusion experiments.

### 2.2. Extraction of Active Compounds

*P. akhurstii* sp. nov. 0813-124 was pre-cultured in a 50 mL conical centrifuge tube containing 15 mL LP medium at 30 °C and 220 rpm for 24 h. *Pre-culture* was subsequently transferred and incubated in a 2 L Erlenmeyer flask containing 500 mL potato dextrose broth (PDB) with 0.5% tryptone at 30 °C and 220 rpm for 4 days. Broth culture was extracted with an equal volume of ethyl acetate (EA) 3 times by mixing vigorously in a 2 L separatory funnel. The organic phase was collected and filtered by ADVANTEC filter paper (Toyo Roshi Kaisha, Tokyo, Japan). After partitioning, the filtrate was concentrated by a vacuum condenser and then lyophilized to powder.

### 2.3. Disk-Diffusion Assay

The disk-diffusion approach was applied to assess the antifungal effects of extracts on *C. gloeosporioides.* Here, 10^5^ CFU of *C. gloeosporioides* spore were seeded on potato dextrose agar (PDA) plates in a Petri dish. A sterilized paper disk around 6 mm in diameter was infused with a volume of 10 μL dissolved extracts in EA (50 mg/mL) on a microscopic slide. EA solvent served as the negative control. After 2 min of standing, paper disks were pressed lightly on the surface of pathogenic plates. Plates with paper disks were then incubated at 30 °C for 3 days. Antifungal activities of extracts were evaluated by the diameter of the inhibition zone generated from paper disks on the spore plate.

### 2.4. Confrontation Assay

*P. akhurstii* sp. nov. 0813-124 was evaluated to suppress mycelial growth of *C. gloeosporioides* using a dual culture technique. Initially, bacterial culture was inoculated into LP broth at 30 °C for 2 days. One active circular fungal plug (5 mm in diameter) from oatmeal agar plates was placed at the center of the PDA plate. Then four of 10 μL bacterial broths were located around the fungal plug at a 2 cm distance. Antagonistic plates were incubated at 30 °C for 4 days to determine the inhibiting interaction.

### 2.5. Medium-Pressure Liquid Chromatography Separation of Crude Extract

The *P. akhurstii* crude extract powder was first dissolved in methanol to generate a high-concentration slurry form in a 10 mL glass vial. Dry silica was added, gradually mixing thoroughly. The mixture was subsequently concentrated by a vacuum condenser to remove methanol entirely and transferred to an empty column. The column containing silica absorbing bacterial crude extract was carefully connected to the top of a commercial silica gel column (SIHP-JP, 50 μm, Interchim, Los Angeles, CA, USA). The column was sequentially eluted with hexane (buffer A), EA (buffer B), and methanol (buffer C), respectively, at a flow rate of 20 mL/min under room temperature using MPLC performed on a Biotage Isolera One (Biotage, Uppsala, Sweden). The composition of the mobile phase is displayed in Appendix A. The maximum of each fraction collected on a 16 mm × 150 mm rack was 20 mL. The fraction mode was controlled by molecular absorption at 254 and 280 nm, concatenated into four fractions as shown in Appendix A. For further experiments, each fraction was concentrated, lyophilized, and stored at −20 °C.

### 2.6. High-Performance Liquid Chromatography Isolation of Medium-Pressure Liquid Chromatography Fractions

Evaluated by disk-diffusion assay, the powders of medium-pressure liquid chromatography (MPLC) elutes of interest were dissolved in methanol and centrifuged at 11,000× *g* for 10 min to remove insoluble particles. The supernatants were further separated and isolated by reverse-phase HPLC using a Hitachi LaChrom Elite system (Hitachi, Tokyo, Japan), consisting of an L-2130 pump and an L-2455 DAD coupled with Alltech 3300 ELSD (Alltech, Deerfield, IL, USA). EZChrom Elite Software for Hitachi Version 317 (Agilent, CA, USA) was applied to collect and analyze data. Samples were loaded on a Discovery HS C18 column (25 cm × 10 mm, 5 μm, Supelco, Cheshire, UK). The mobile-phase components were water +0.1% trifluoroacetic acid (Buffer A) and acetonitrile (ACN) (Buffer B). The peak detection was monitored by absorption wavelength at 260 nm. The column was eluted at a flow rate of 2.5 mL/min. Forty-five one-minute fractions were collected by auto-collector based on retention time over 1–45 min intervals. The setting program of the mobile phase was displayed in Appendix A. For further experiments, fractions were concentrated, lyophilized, and stored at −20 °C.

### 2.7. Compound Prediction Using LC-MS/MS

The freeze-dried HPLC elutes possessing antifungal activity were dissolved in methanol for LC-MS/MS analysis. The C18 column (Waters UPLC ACQUITY UPLC BEH C18 1.8 μm, 100 mm × 2.1 mm, Waters, Milford, MA, USA) was used with the following gradients: 0–6 min at 5–99.5% of ACN + 0.1% formic acid, 6–8 min at 99.5% of ACN + 0.1% formic acid, 8–8.2 min at 99.5%–5% of ACN + 0.1% formic acid and 8.2–10 min at 5% of ACN + 0.1% formic acid. The rate of flow was 0.4 mL/min. The mass data were analyzed by the Thermo Orbitrap Elite system, and the profile mode was set to positive mode with a mass range of *m/z* 50–1500. The resolution of Orbitrap was 60,000 resolution (MS1), 15,000 (DDA), and 30,000 (DDA2). For tandem mass (MS/MS), the top five intense ions from each full mass scan were selected for collision-induced dissociation fragmentation (CID). Then the fragments from MS/MS data were used to predict the structures.

### 2.8. NMR Analysis of Isolated Peaks from HPLC

^1^H, HSQC, HMBC, and COSY NMR spectra were recorded with Bruker 600 MHz Spectrometer with an ultra-shielded Plus magnet (HFNMRC, Academia Sinica). Topspin software (Bruker, Billerica, MA, USA) was used to collect and analyze data. Deuterated NMR (DMSO-d_6_) solvents were purchased from Cambridge Isotope (Cambridge, MA, USA). A 3 mm Shigemi Tube Set (10 mm Bottom, DMSO-d_6_, Universal, Shigemi, Tokyo, Japan) was applied to detect rare samples. The structures were unveiled through 1D and 2D NMR experiments (Appendix A). The chemical shifts of 1D spectra were calibrated from the DMSO-d_6_ solvent peak (256 K) at δ_H_ 2.50.

Cepafungin I: white powder; HRMS(ESI+) *m/z* 535.3482 [M+H]^+^ (calcd. for C_28_H_46_N_4_O_6_, 534.3417); NMR data (Appendix A) were identical with reported data in [29].

Glidobactin A: white powder; HRMS(ESI+) *m/z* 521.3328 [M+H]^+^ (calcd. for C_27_H_44_N_4_O_6_, 520.3261); NMR data (Appendix A) were identical with reported data in [29].

### 2.9. Determination of Minimum Inhibitory Concentration

An agar dilution method was employed to determine the minimum inhibitory concentrations (MICs) of purified compounds, PL1 crude EA extract, and carbendazim served as the positive control against *C. gloeosporioides* in a sterilized 24-well plate. Treatments (20 μL) dissolved in methanol were added in 380 μL molten PDA at 55 °C to obtain five final concentrations of 2.5, 1.25, 0.625, 0.3125, and 0.15625 μg/mL for carbendazim; 300, 250, 200, 150, 100, and 50 μg/mL for PL1 crude EA extract; 3, 2.5, 2, 1.5, 1, and 0.5 μg/mL for two isolated compounds. Molten medium amended with treatment was then quickly loaded in each well. The PDA containing 5% methanol by volume was the negative control. Subsequently, 10 μL *C. gloeosporioides* spore suspension containing 1000 CFU was dropped on the surface of PDA in a 24-well plate. The plate was sealed and incubated at 30 °C for 3 days. MIC was determined as the minimum concentration of samples that exhibited the ability to suppress the germination of *C. gloeosporioides* spore fully.

### 2.10. RNA Isolation, Reverse Transcription, and Quantitative Reverse Transcription PCR

For RNA extraction of bacterial culture in PDB, 2 mL of culture broth was centrifuged for 10 min at 8000× *g* at 4 °C to obtain bacterial pellets. Total RNAs were harvested by mixing thoroughly with 500 μL TRIZOL (Ambion, MA, USA). After 5 min of standing, samples were mixed and shaken thoroughly with 100 μL chloroform, about 1/5 of TRIZOL volume, and then left to stand for 3 min at room temperature. Next, samples were centrifuged for 15 min at 12,000× *g* at 4 °C, 250 μL isopropanol was added to the supernatants and placed at room temperature for 5 min to deposit the RNAs. After 12,000× *g* centrifugation for 10 min at 4 °C, the pellets were washed with 1000 μL 75% ethanol three times and 99% ethanol once by mixing mildly and centrifuging for 5 min at 12,000× *g* at 4 °C. Finally, the supernatants were removed to obtain pellets, and pellets were air-dried for 20 min. The RNAs in 20 μL RNase-free water were qualified using a NanoDrop 1000 Spectrophotometer (Thermo Fisher Scientific, MA, USA).

After qualification, ToolsQuant II Fast RT Kit (BioTools, New Taipei City, Taiwan) was applied to generate cDNA. Briefly, a gDNA removal reaction containing 8 μL bacterial RNA (2000 ng) and 2 μL 5× gDNA Eraser was incubated at 42 °C for 3 min. Then, 2 μL 10× Fast RT Buffer, 1 μL RT Enzyme Mix, 2 μL RT Primer Mix, and 5 μL RNase-Free ddH_2_O were added to the gDNA removal reaction. The sample mixture was mixed fully and incubated at 42 °C for 15 min. After incubation at 95 °C for 3 min, synthesized cDNA was immediately stored at −20 °C for further quantitative PCR experiments.

For qRT-PCR, 2 μL cDNA was mixed with 7.2 μL ddH_2_O, and then 0.4 μL of forward and reverse specific primers (100 μM), 10 μL SYBR Green Master Mix was added. The primer sets were provided in Appendix A. After mixing well, qRT-PCR was conducted using a CFX96 Real-Time PCR Detection System with C1000 thermal cycler (Bio-Rad, Hercules, CA, USA). The program set up was 5 min at 95 °C first and then 39 cycles of 10 s at 95 °C, 30 s at 60 °C, and 30 s at 72 °C. The fortieth cycle was 15 s at 95 °C, 5 s at 60 °C, and 0.5 °C gradual ascension up to 95 °C to confirm the specificity of primers. The data were analyzed with the CFX Manager software, and the CT (cycle of threshold) was detected to determine the expression level of genes. The CT value of the target gene was further normalized to the 16S gene, which was used as the reference gene. Finally, normalized gene expression was calculated with the 2-ΔΔCt method. In terms of time-course qRT-PCR, the significant difference compared to the time point of 0 h was determined using *t*-test.

### 2.11. Biosynthetic Gene Clusters Analysis

The antiSMASH web server: https://antismash-db.secondarymetabolites.org/ (accessed on 6 May 2020) [30] was used to predict biosynthetic gene clusters encoding diverse secondary metabolites from the complete genome of *P. akhurstii* sp. nov. 0813-124 phase I (GenBank Accession Number: CP022160.1) [30].

## 3. Results and Discussion

### 3.1. Phenotypic Comparison of PL1 and PL2; Appearance and Antifungal Activities

The phenotypes of PL1 and PL2 differed in pigmentation and antifungal capabilities. This phenotypic switch can also be observed after prolonged cultivation under laboratory conditions [31]. After three days of culture, PL1 turned dark brown, whereas PL2 became white or pale, suggesting diverse secondary metabolite production between the two phases of *P. akhurstii* sp. nov. 0813-124 (Figure 1a,b). Several studies have reported similar results for different pigmentation types [23,32,33]. The main factor causing the dark red color was the yield of anthraquinones [34]. The anthraquinone-related operon *antABCDEFGHI* activated through a transcriptional regulator, AntJ, corresponds to the type II PKS and enzymes for anthraquinone biosynthesis [35,36]. 

In recent decades, the main studies on *P. akhurstii* or *Photorhabdus* have been in the insecticide field because *Photorhabdus* was initially isolated from the nematodes. Recently, many studies have explored whether *Photorhabdus* could serve as a potential biocontrol against pathogenic fungi or bacteria [12,16,25,37]. Therefore, we investigated the antifungal activity of PL1 and PL2 against *C. gloeosporioides*. The results showed that PL1 suppressed *C. gloeosporioides* mycelial growth and spore germination, leading to a clear inhibition zone (Figure 1c,e), but PL2 did not affect *C. gloeosporioides* spread (Figure 1d,f). This result was similar to previous studies showing that the secondary variant did not demonstrate antimicrobial properties [31].

### 3.2. Analysis of Bacterial Extracts

PL1 EA crude extract was separated into four fractions eluted with gradient solvent made of hexane, EA, and methanol by MPLC (Appendix A). The fourth MPLC eluent exhibited the most potent inhibitory effect on *C. gloeosporioides* spore germination (Figure 2a). Advanced compound isolation was carried out by HPLC, and the inhibition capability of the 45 one-minute subfractions was determined using an antifungal bioassay. The bioassay results indicated that subfractions 17, 18, 22, 23, 27, and 28 could suppress the *C. gloeosporioides* spore germination (Figure 2b). A total of five active peaks were collected from the subfractions mentioned above: Peak 1 was the most abundant and corresponded with subfractions 17 and 18. Peaks 2, 3, 4, and 5 were mainly from subfractions 22, 23, 27, and 28, respectively (Figure 2c).

Further identified by MS/MS fragmentation, the structures of all peaks were proposed to be glidobactins or cepafungins (Figure 3). Because of the low mass of peaks obtained (Appendix A), we putatively identified Peak 3 as glidobactin C based on MS/MS fragmentation. On the other hand, Peaks 1 and 2 were sufficient to be assigned as glidobactin A and cepafungin I by NMR data and MS/MS fragmentation, respectively [38]. Idiosyncratically, PL1 was triggered to accumulate cepafungin I when incubated on PDA compared to LP agar or PDB. However, PL2 produced little glidobactin A on PDA and barely produced any bioactive compounds in the other media (Appendix A). This is the first study to report that glidobactin A and its analog, cepafungin I and glidobactin C, are the major antifungal ingredients from the EA crude extract of *P. akhurstii*. Glidobactin A, cepafungin I, and glidobactin C have been reported to have antitumor and antimicrobial characteristics [39,40]. The glidobactins are well known for being inhibitors of the eukaryotic 20S proteasome. The mechanism of glidobactin-like inhibitors might result from their specific structure interfering with the 20S proteasome [38]. Glidobactins, cepafungins, and syringolins, identified as the syrbactin natural product class [41], bear a 12-membered dipeptide-macrolactam with an α,β-unsaturated amide, giving these compounds designated chemical and biological functionality [42]. Given an α,β-unsaturated carbonyl group, glidobactins and cepafungins potently react with the catalytic sites of the β2 and β5 subunits of the eukaryotic 20S proteasome to block and inhibit trypsin-like and chymotrypsin-like activities, respectively, through the irreversibly covalent bond between the α,β-unsaturated moiety and the hydroxy group of a catalytic threonine residue at low concentrations [41]. Therefore, due to the advantage of cytotoxicity, glidobactins and cepafungins have been widely studied for medical use as therapeutic strategies with antiproliferative effects in several tumor cell types [43]. Moreover, instead of the substantial adverse effects of clinical drugs, glidobactin A, cepafungin I, and glidobactin C might be more selective for the immunoproteasome induced during disease processes [44].

### 3.3. Minimum Inhibitory Concentration

The antifungal activities of the extract and isolated compounds from PL1 are summarized in Table 1. The minimum inhibitory concentration (MIC) value was determined using the agar dilution method and compared with carbendazim as the standard antibiotic in 24-well culture plates with three biological repeats (Appendix A). The crude butanol extract did not show any antifungal effect on *C. gloeosporioides*, but the crude EA extract exhibited antifungal ability with a MIC lower than 300 µg/mL. In addition, the MIC values of glidobactin A and cepafungin I against *C. gloeosporioides* were lower than 1.5 and 2.0 µg/mL, respectively. Although glidobactins have been reported to have a broad antifungal ability against phytopathogens [39,45,46], only the analog syringolin has been applied to control agricultural rice blast disease caused by *Pyricularia oryzae* [47]. Relevant studies have scarcely touched upon the effect of glidobactins in economic crops by field trials. Furthermore, few studies have explored the possibility of an application using glidobactins as a biocontrol agent that might function as an inhibitor of the eukaryotic proteasome mentioned above. Nevertheless, it has been observed that plants are not essentially sensitive to glidobactin A, which implies that it might not be taken up by plants completely [48]. In addition, the potency of glidobactin against *C. gloeosporioides* in this study can be compared with another study indicating that the MIC values of pesticide products, including fludioxonil, cyprodinil, and iprodione, against *C. gloeosporioides* from strawberry, were all higher than 10 mg/L [6].

### 3.4. Genomic Analysis and Expression Level of the Glidobactin BGC in PL1 and PL2

We explored the BGCs of *P. akhurstii* sp. nov. 0813-124 phase I using antiSMASH 5.1 [30] to predict the region of BGC corroborating glidobactins in bacterial secondary metabolites. The genomic analysis presented 22 BGCs, which mostly encoded NRPS or PKS types of BGCs, and implied that the 8th and 15th BGCs were most likely to correspond to luminmycin (100% similarity) and glidobactin (15% similarity), respectively. In addition, the 8th BGCs showed another result producing glidobactins with 26% similarity as well (Appendix A). By confirming the functions of the NRPS biosynthetic genes, it was revealed that the 15th BGC was not responsible for glidobactins, albeit being partially similar. It has been demonstrated that the glidobactin BGC made of eight genes from *glbA* to *glbH* was first found in *Schlegelella brevitalea* sp. nov. DSM 7029 [46]. The core biosynthetic genes are *glbC* and *glbF*, which encode the hybrid NRPS-PKS and NRPS modules constituting the macrolactam part of glidobactins (Appendix A). Intriguingly, *glbB*, which is not one of the core biosynthetic genes, encodes a lysine 4-hydroxylase acting as a rate-limiting agent in the biosynthesis of glidobactins, responsible for catalyzing the 4-hydroxylation reaction of L-lysine [46,49]. Only the 8th BGC encoded the core biosynthetic units of the glidobactin BGC in *P. akhurstii* sp. nov. 0813-124. The 8th BGC shared 26% similarity containing *09515*, *09520*, *09525*, *09530*, and *09535* genes corresponding to *glbG*, *glbF*, *glbD*, *glbC*, and *glbB*, respectively (Figure 4).

To investigate the expression level of the glidobactin BGC in PL1 and PL2, we further measured the gene expression corresponding to the glidobactin BGC during a time-course incubation (Figure 5a–e). The expression profiles of these genes peaked earlier in PL1 than PL2 after transfer to PDB. In PL1, genes involved in glidobactin biosynthesis generally exhibited rapid expression within 24 h of incubation and returned to the control level at 30 h; in PL2, the expression levels were stimulated after 60 h. Interestingly, although most studies indicated that the secondary variant would not produce bioactive compounds [20,21], we found that PL2 still secreted a little glidobactin A on PDA (Appendix A). This phenomenon was consistent with the expression of glidobactin BGC of PL2 during the post-incubation phase. Compared with PL1, the glidobactin BGC expression of PL2 peaked late with a large deviation after 60 h of incubation. This phenotypic heterogeneity between PL1 and PL2 might be elucidated through the bacterial life cycle implicated in the developing stage of the *Heterorhabditis* nematode host. Initially, only the primary variants of bacteria replicate preferentially within the anterior intestinal tracts of juvenile infected nematodes [21]. During the infection in which nematodes parasitize insects or larvae, the primary variants of *Photorhabdus* are released into the insect hemolymph and grow exponentially. The primary variants can support the growth and development of nematodes by secreting toxins and exoenzymes to bioconvert the insect cadaver into a nutritious source.

During the prolonged stationary period in the insect cadaver, a vast subset of *Photorhabdus* switched from primary to secondary variants due to depleted nutrients from the insect cadaver [20]. In contrast to primary variants, secondary variants of *Photorhabdus* were less efficacious at providing conditions to assist nematode growth and development. In addition, it was revealed that several stress-related genes and proteins were downregulated in the secondary variant of *Photorhabdus* [23]. Instead of producing toxins, exoenzymes, or other antimicrobial substances, the energy metabolism of secondary variants is more robust to cope with challenging environmental conditions, including starvation or osmotic stress [20,21].

## 4. Conclusions

In the present study, *P. akhurstii* sp. nov. 0813-124 effectively hindered the spread of the mycelial growth of *C. gloeosporioides*. Meanwhile, the EA crude extract of PL1 exhibited superior inhibition against the fungal spore germination. Three components, glidobactin A, cepafungin I, and glidobactin C, were identified in the active fractions of the PL1 crude extract. After determining the MICs, glidobactin A and cepafungin I showed profound antifungal activity against *C. gloeosporioides* at low concentrations. To evaluate the phase variation between different types of *P. akhurstii* sp. nov. 0813-124, we measured the gene expression of glidobactin BGC in both PL1 and PL2 during a time-course incubation, demonstrating that the primary form of *P. akhurstii* sp. nov. 0813-124 had a faster reaction to facilitate the production of bioactive compounds than the secondary variant, which might be attributed to the strong antimycotic effect of PL1, although PL2 also secreted a small amount of glidobactin A at late time points. Our findings indicate that *P. akhurstii* sp. nov. 0813-124 and its metabolites have great potential as novel biological agents to control the invasion of phytopathogens.

## Figures and Tables

**Figure 1 jof-08-00403-f001:**
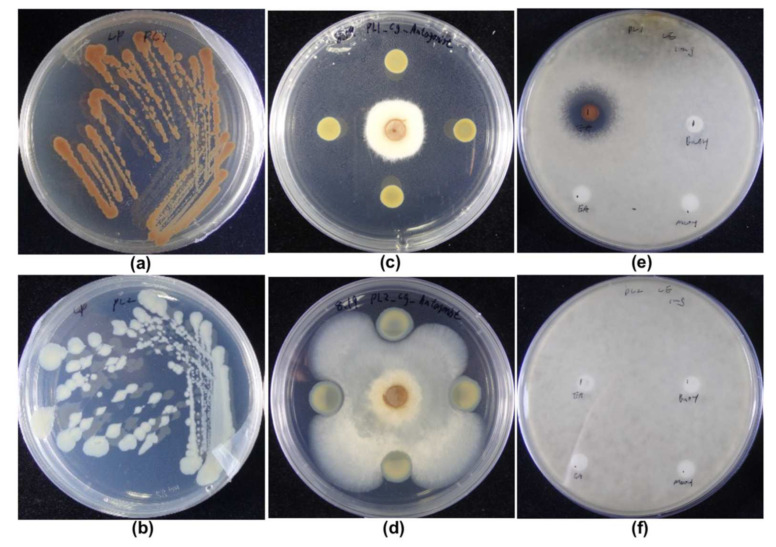
Fungal inhibition against *C. gloeosporioides* and pigmentation of PL1 and PL2. Pigmentation of PL1 (**a**) and PL2 (**b**) after 72 h incubation at 30 °C on LP agar. The antagonist assays of PL1 (**c**) and PL2 (**d**) against *C. gloeosporioides* plugin (center). The inhibition of fungal spore germination by EA solvent (left bottom) and crude extracts of PL1 (**e**) and PL2 (**f**) using methanol (right bottom), butanol (right top), and EA (left top).

**Figure 2 jof-08-00403-f002:**
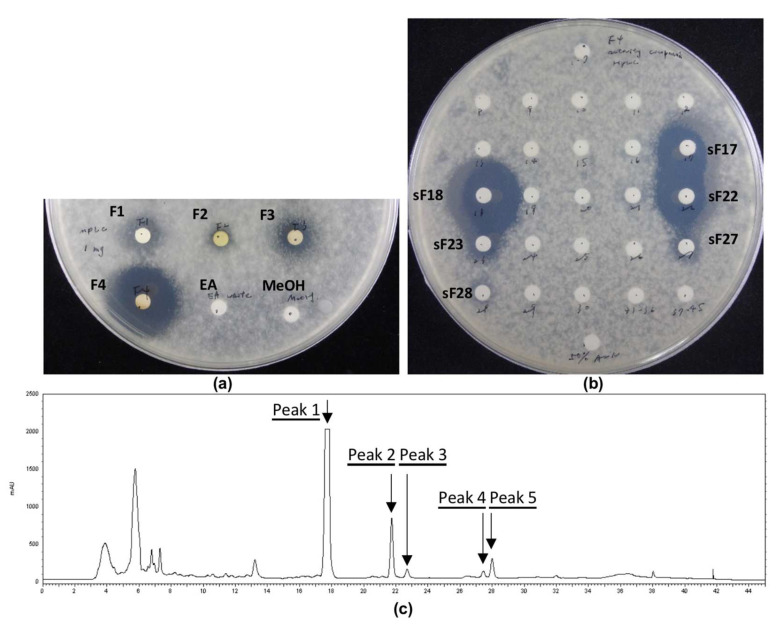
Growth inhibition of *C. gloeosporioides* spores by eluents from PL1 EA crude extract on PDA. (**a**) Four fractions of MPLC eluents. (**b**) Forty-five subfractions of HPLC eluents. (**c**) HPLC profile of the MPLC eluent F4.

**Figure 3 jof-08-00403-f003:**
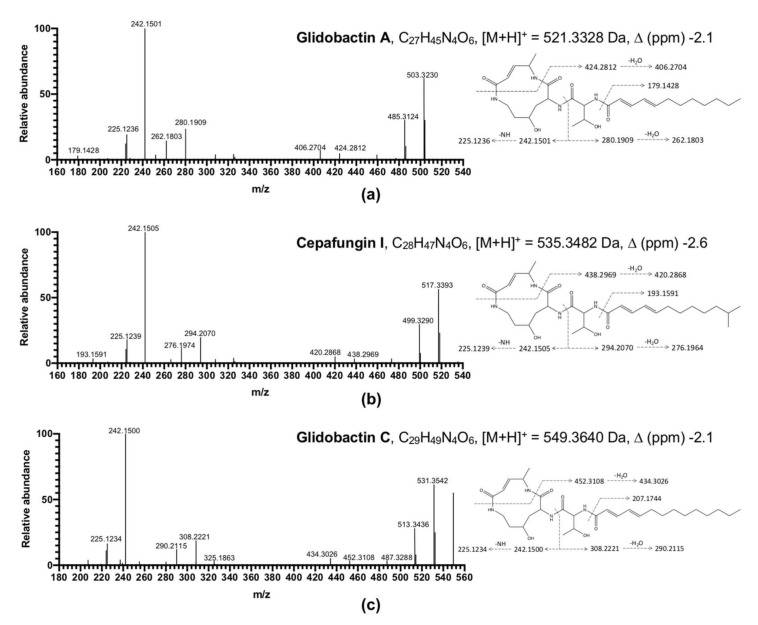
Structures and MS^2^ fragment annotation of HPLC Peak 1 (**a**), Peak 2 (**b**), and Peak 3 (**c**).

**Figure 4 jof-08-00403-f004:**
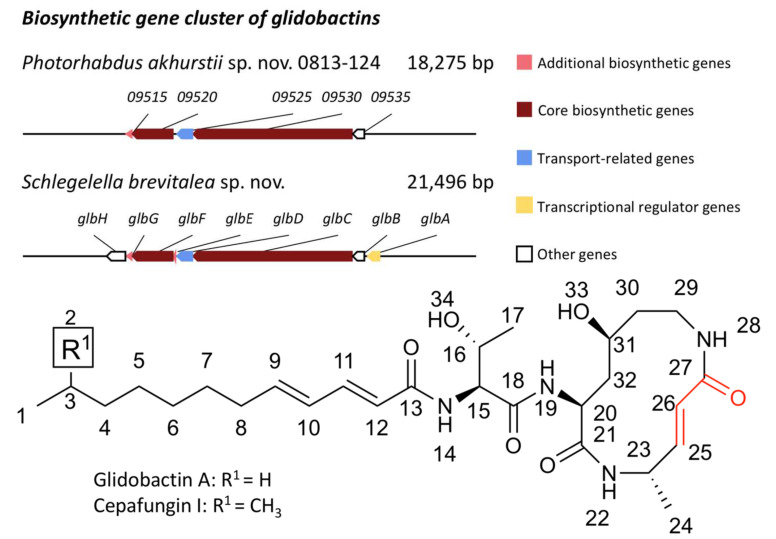
The 8th BGC of *P. akhurstii* sp. nov. 0813-124 phase I and reference glidobactin BGC from *S.*
*brevitalea* sp. nov. (MIBiG accession: BGC0000997). The structures of glidobactins are shown at the bottom. The functional carbonyl group is labeled in red.

**Figure 5 jof-08-00403-f005:**
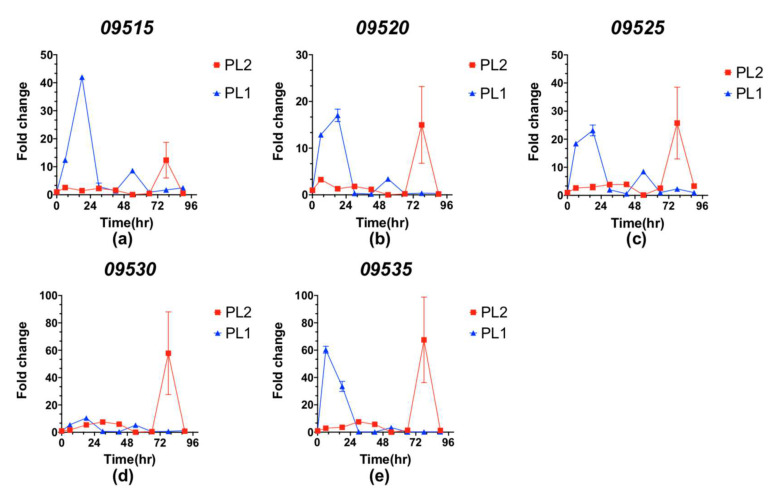
Time-dependent expressions of five related genes (**a**–**e**) in the 8th BGC compared between PL1 (blue) and PL2 (red) during 90 h of PDB incubation. Values are the means of biological replicates ± standard deviations (*N* = 3).

**Table 1 jof-08-00403-t001:** Inhibition bioassay and minimum inhibitory concentrations (MIC) of PL1 extract and isolated compounds against *C. gloeosporioides* spore.

Subject	MIC (µg/mL)
PL1 BuOH crude extract	NS *
PL1 EA crude extract	<300
Glidobactin A	≤1.50
Cepafungin I	≤2.00
Carbendazim (positive control)	≤0.32

* NS: non-sensitive.

## Data Availability

Data is contained within the article and Appendix A.

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
