# Peer review of "Evaluation of the Antifungal Activities of Photorhabdus akhurstii and Its Secondary Metabolites against Phytopathogenic Colletotrichum gloeosporioides"

_jof, 2022, doi:10.3390/jof8040403_

Round 1
Reviewer 1 Report
The manuscript entitled “Evaluation of the antifungal activities of Photorhabdus akhurstii and its secondary metabolites against phytopathogenic Colletotrichum gloeosporioides” feature the antifungal activity of the extract in different phases of the fungal growth, the isolation and identification of the bioactive ingredients responsible of that activity and demonstrating the potential use as a biocontrol agent against C. gloeosporioides. I think its suitable for publication but there are some questions the authors must address:
- How did the authors to assure the identity of gloeosporioides? And from where they obtained the strawberries samples? Are commercially acquired?
- The authors did not mention any mass quantity obtained in the extraction process neither in the fractionation or the obtained compounds. In each step, they used all the available material.
- Section 2.4 must be improved that lectors can understand the identity and position of each fungal plug. I have to go to the figure to figure out that. Is the concentration of the extract equal in the disk diffusion assay and in the confrontation assay?
- Indicate how you based to get the four fractions of the crude extract
- Indicate how many fractions resulted from fractionation of HPLC F4 and how you based to group the fractions. In section 3.2 you refer as peaks 1-5 and then you showed F17-18, F22-23, F27-F28 in figure 2. That may be confusion.
- In Line 154 must be Waters instead of Water
- In section 2.7 must clarify the composition of the elusion gradient because you are mentioning just the AcCN + 0.1% FA. What FA stand for?
- Did you measure the optical rotation of the isolated compounds?
- You stated that only could obtain sufficient mass to characterize glidobactin A and cepafungin I but in figure 3 and figure S7 its clear to me that you were able to identify glidobactin C at least by LC-MS and LC-MS/MS in the active fraction. So why you let the glidobactin C out of your results and discussion? There are previous reports about any activity of glidobactin C? How does fit the presence of glidobactin C in your results?
- If you were able to run HSQC and HMBC experiments, why you miss to present the 13C- NMR spectra in the supplementary info.
Reviewer 2 Report
Reviewer comments
Manuscript title: Evaluation of the antifungal activities of Photorhabdus akhurstii and its secondary metabolites against phytopathogenic Colletotrichum gloeosporioides
The author studied the antimicrobial activity of Photorhabdus akhurstii against C. gloeosporioides. They identified the bioactive ingredients. Also, they revealed the biosynthetic gene cluster of corresponding bioactive molecules through genomics analysis and determined its expression levels in two variants of Photorhabdus akhurstii.
Comments
- The English usage need revision.
- This study lack the novelty. It is known that Photorhabdus have antimicrobial effect against Colletotrichum gloeosporioides.
- The author in page 7 line 274-275 claimed that This is the first study in which glidobactin A and its analog cepafungin I, have been found to be the major bioactive ingredients from the extract of P. akhurstii. This is not true. It was reported before. As example check (Characterization of active compounds against strawberry anthracnose produced by Photorhabdus akhurstii (2019)). DOI: 6342/NTU201902164 .
- In the literature, there are a large numbers of researches about the biosynthetic gene cluster of corresponding bioactive molecules through genomics analysis in Photorhabdus.
Therefore, my opinion is this manuscript is not suitable for publication in J Fungi.
Round 2
Reviewer 1 Report
Thanks to the authors for considering my comments into account and resolving them in the best possible way. I feel that the understanding of the work has greatly improved.
But it seems that in section 3.2, the authors still not specified the presence/isolation of glidobactin C.
I could suggest to re-write lines 270-273, indicating which peak in Figure 2 corresponds to glidobactin C and then specify that because of the low mass obtained it was putatively identified by MS/MS fragmentation in addition to the identification/characterization of the glidobactin A and cepafungin I by NMR data and MS/MS fragmentation.
And then, in the conclusions section, add/mention that glidobactin C was also a component of PL1 extract.
After that i think the manuscript will be ready for publication.
Reviewer 2 Report
The manuscript is improved and It is suitable for publication in the present form.
Author Response
Thank you.